# Design and Fabrication of Silicon Pressure Sensors Based on Wet Etching Technology

**DOI:** 10.3390/mi16050516

**Published:** 2025-04-28

**Authors:** Fengchao Li, Shijin Yan, Cheng Lei, Dandan Wang, Xi Wei, Jiangang Yu, Yongwei Li, Pengfei Ji, Qiulin Tan, Ting Liang

**Affiliations:** 1State Key Laboratory of Extreme Environment Optoelectronic Dynamic Measurement Technology and Instrument, North University of China, Taiyuan 030051, China; lifengchao@nuc.edu.cn (F.L.); s202206079@st.nuc.edu.cn (S.Y.); wdd15503435005@163.com (D.W.); 13754766159@163.com (X.W.); yujg@nuc.edu.cn (J.Y.); flyrubbit@163.com (P.J.); tanqiulin@nuc.edu.cn (Q.T.); 2Department of Automation, Taiyuan Institute of Technology, Taiyuan 030051, China; liyw@tit.edu.cn

**Keywords:** silicon, piezoresistive, pressure sensor, wet etching, sensitive membrane

## Abstract

This paper presents a novel silicon-based piezoresistive pressure sensor composed of a silicon layer with sensing elements and a glass cover for hermetic packaging. Unlike conventional designs, this study employs numerical simulation to analyze the influence of varying roughness levels of the sensitive membrane on the sensor’s output response. Simulation results demonstrate that pressure sensors with smoother sensitive membranes exhibit superior performance in terms of sensitivity (5.07 mV/V/MPa), linearity (0.67% FS), hysteresis (0.88% FS), and repeatability (0.75% FS). Furthermore, an optimized process for controlling membrane roughness was achieved by adjusting the concentration of the etchant solution. Experimental results reveal that a membrane roughness of 35.37 nm was attained under conditions of 80 °C and 25 wt% TMAH. Additionally, the fabrication process of this piezoresistive pressure sensor was significantly simplified and cost-effective due to the adoption of a backside wet etching technique. The fabricated sensor demonstrates excellent performance metrics, including a sensitivity of 5.07 mV/V/MPa, a full-scale (FS) output of 101.42 mV, a hysteresis of 0.88% FS, a repeatability of 0.75% FS, and a nonlinearity of 0.67% FS. These results indicate that the proposed sensor is a promising tool for precise pressure measurement applications, offering both high performance and cost efficiency. This study not only advances the understanding of the impact of membrane roughness on sensor performance but also provides a practical and scalable fabrication approach for piezoresistive pressure sensors.

## 1. Introduction

Pressure sensors are critical components in modern industries [1], medical electronics, aerospace [2], and automotive electronics [3], where their performance directly impacts the measurement accuracy and reliability of systems. Among various types, silicon piezoresistive pressure sensors have emerged as a leading solution for high-precision pressure measurement due to their high sensitivity [4], strong stability, ease of miniaturization, and compatibility with integrated circuit (IC) fabrication processes. However, traditional silicon piezoresistive sensors predominantly rely on dry etching techniques [5], such as reactive ion etching (RIE), to form sensitive diaphragm structures [6]. These methods are associated with high costs, complex equipment, and potential plasma-induced degradation of the piezoresistive layer’s electrical properties, which hinder large-scale production and performance optimization.

In recent years, wet etching technology has regained attention in microelectromechanical systems (MEMS) fabrication due to its low cost, high controllability, and minimal lattice damage [7,8]. This technique employs chemical solutions to anisotropically or isotropically etch silicon wafers, enabling the precise formation of complex three-dimensional microstructures, such as cup-shaped cavities or cantilever beams [9]. This provides new avenues for optimizing diaphragm design and stress distribution in silicon piezoresistive sensors. For instance, wet etching of single-crystal silicon using KOH or TMAH solutions leverages crystallographic plane-selective etching to achieve highly uniform thin diaphragms, thereby enhancing sensor sensitivity and linearity [7,10,11]. Nevertheless, the precision of wet etching processes is influenced by multiple coupled factors, including solution concentration, temperature, etching time, and mask design [12]. Achieving high consistency and low-defect sensitive structures through process optimization remains a significant challenge.

Several studies have reported on pressure sensors fabricated using wet etching techniques. Vinod Belwanshi et al. [6] utilized TMAH wet etching to create a sensitive diaphragm, achieving a sensor sensitivity of 0.147 mV/V/bar at room temperature. Jun-Hwan Choi et al. [13] developed a hydrogen pressure sensor on SOI wafers using wet etching, reporting maximum nonlinearity, hysteresis, temperature coefficient of resistance (TCR), and sensitivity values of −0.341%, 0.909%, 4128 ppm/°C, and 34.22 mV/V, respectively. The XTEH-10LAC-190 series pressure sensor by Kulite Corporation employs wet etching to precisely form an E-type sensitive diaphragm, enabling stable operation at high temperatures up to 482 °C [14]. This sensor, with a test pressure range of 0.35 MPa to 7 MPa, exhibits static performance metrics better than 0.25% FS, with thermal zero drift and thermal sensitivity drift controlled within ±1.5% FS/100°F.

This study addresses the aforementioned challenges by proposing an innovative design and fabrication method for silicon piezoresistive pressure sensors based on wet etching technology. To improve the accuracy of simulation models in predicting sensor performance, we consider the primary influences of process errors on device structure, including diaphragm thickness and piezoresistor positioning. By combining simulation analysis with experimental validation, we enhance the predictive accuracy of the simulation model. First, finite element analysis (FEA) is employed to investigate the impact of varying diaphragm thicknesses on piezoresistive stress distribution, optimizing the mechanical performance of sensitive elements. Second, a systematic study is conducted to establish the relationship between etchant concentration, silicon etching rate, and surface morphology, defining a high-precision process window for diaphragm fabrication. Finally, a high-performance silicon piezoresistive pressure sensor is fabricated by integrating doping, piezoresistor etching, wet etching, and anodic bonding processes. The sensor’s performance is validated through pressure-electrical characteristic testing, including sensitivity, hysteresis, repeatability, and nonlinearity.

This research aims to provide theoretical and experimental foundations for the development of low-cost, high-reliability silicon-based pressure sensors, advancing the application of wet etching technology in MEMS manufacturing. The proposed methodology not only addresses current limitations in sensor fabrication but also paves the way for future innovations in the field.

## 2. Theory, Design, and Simulation

The operating principle of piezoresistive pressure sensors is based on the piezoresistive effect of silicon materials, which is a change in the resistivity of a semiconductor material in response to an external force. The schematic diagram of the pressure sensor is shown in Figure 1 [4]. Four silicon piezoresistive resistors were fabricated on the front side of the sensor-sensitive chip to form a Wheatstone bridge circuit. The back side was formed by a wet etching technique to form the sensitive diaphragm of the pressure sensor. The monolithic silicon is bonded to a glass cover to form an adiabatic cavity.

The equivalent circuit of the Wheatstone bridge is shown in Figure 2. The equivalent circuit of the Wheatstone bridge is shown in Figure 2. The total resistance Rz is obtained by the arithmetic addition of the components R_1_, R_2_, R_3_, and R_4_ as followsRz = R_1_ + R_2_ + R_3_ + R_4_.(1)

When pressure is applied, the pressure-sensitive diaphragm undergoes mechanical deformation, causing the resistor located on the diaphragm to deform accordingly and its resistance to change. In this case, R_1_ and R_3_ are located in the tensile direction and decrease in resistance, while R_2_ and R_4_ are located in the compressive direction and increase in resistance [4,15]. This symmetrical change makes the output voltage of the bridge linearly related to the external pressure, thus realizing accurate measurements.

If the resistance value of each resistor changes by Δ*R*, when the input voltage *V_in_* is constant, the output voltage *V_out_* can be expressed as follows:(2)$$V_{out}=V_{0}+\frac{\Delta R}{R}V_{in},$$
where *V_in_* is the excitation voltage and Δ*R* is the resistance change.

The operational principle of piezoresistive pressure sensors is fundamentally based on the deformation of the sensitive membrane under applied pressure. This deformation subsequently induces an imbalance in the Wheatstone bridge circuit, thereby altering the output signal, which adheres to the theoretical model of membrane deformation under minimal deflection [16,17]. To achieve a highly linear output from the sensor, it is generally stipulated that the maximum allowable deformation should not exceed 20% of the membrane thickness [1]. Within this range, the membrane undergoes bending without stretching. To meet the overload resistance requirements of the sensitive membrane, the maximum stress on the membrane surface should be less than one-fifth of the silicon fracture stress [10]. The full-scale output principle ensures that the electrical connections of the chip are not compromised by excessive current. Taking these factors into account, the design requirements are expressed in Equations (2) to (4) as follows:(3)$$\omega_{\max}=\frac{0.0138Pa^{4}}{Eh^{3}}\leq \frac{h}{5},$$(4)$$\sigma_{\max}=\frac{0.308Pa^{2}}{h^{2}}\leq \frac{\sigma_{m}}{n},$$(5)$$V_{out}=\frac{68.1\times 10^{-11}Pa^{-1}(A-B)}{2+68.1\times 10^{-11}Pa^{-1}(A+B)}V_{in}$$

In the formula, *ω*_max_ is the maximum deflection, E is the Young’s modulus of silicon, *A* is the average stress parallel to the edge of the film, *B* is the average stress perpendicular to the edge of the film, h is the thickness of the film, *V_in_* is the input voltage, and *V_out_* is the full-scale output.

Additionally, *σ*_m_ = 6 GPa is the ultimate strength of monocrystalline silicon, and *n* is the safety factor (*n* = 5). The parameters *A* and *B* represent the average stresses on the piezoresistor parallel and perpendicular to the edges of the sensitive diaphragm, respectively.

A 3D intercept line with a length of 1 mm is set at the position of the center axis of the sensitive diaphragm, as shown in Figure 3a, and another 3D intercept line along the edge of the sensitive diaphragm, as shown in Figure 3b, is set. The width and length of the stress concentration zone generated at the edge of the sensitive diaphragm can be obtained by two path analyses, as shown in Figure 3c.

Taking into account the process nesting alignment accuracy and improving the process redundancy, the piezoresistive pressure sensor designed in this paper will lay the piezoresistor structure at a position 20 μm away from the edge of the diaphragm.

In the solid mechanics interface settings, the chip bottom was defined as a fixed constraint boundary, considering the chip is mounted in a face-up configuration. A pressure load of 4 MPa was applied to the etched bottom surface. In the parametric scanning simulation, the membrane thickness was varied from 86 μm to 100 μm in 1 μm increments starting from the theoretically calculated minimum thickness as the lower limit.

The relationship between the full-scale output and the membrane thickness is illustrated in Figure 4. The simulation results demonstrate that as the membrane thickness increases, the full-scale output, or sensitivity, decreases monotonically. At a membrane thickness of 95 μm, the full-scale output is approximately 101.03 mV, corresponding to a sensitivity of about 5.0515 mV/V/MPa.

Based on the above analysis and process feasibility, the size of pressure-sensitive diaphragm is designed as 1000 μm × 1000 μm × 95 μm, and the overall size of the pressure sensor chip is 3200 μm × 3200 μm × 1000 μm.

## 3. Fabrication

### 3.1. Material and Characterization

#### 3.1.1. Etching Rate and Roughness Analysis

As shown in Figure 5, silicon crystallizes in a diamond cubic structure and exhibits different crystallographic orientations, mainly on the (100), (110), and (111) faces [7]. These planes demonstrate significant variations in atomic arrangement and density, with the (111) plane exhibiting the most compact atomic packing and the (100) plane showing relatively sparse atomic distribution. The differential chemical bond density and bond energy across these planes result in anisotropic chemical reactivity when exposed to specific etchants. This crystallographic-dependent reactivity leads to orientation-dependent etching rates, where the (100) and (110) planes typically exhibit substantially higher etching rates compared to the (111) plane [8,15,18]. This difference in etching rates causes the etching front to gradually converge toward the [19] plane. Given the fixed angle of 54.7° between the planes, the process ultimately forms inclined sidewalls with this characteristic angle [20]. This property is of significant importance in applications such as MEMS and semiconductor manufacturing [21].

Based on the mechanism of how crystal orientation affects etching behavior, this study further explores the regulation of surface morphology by process parameters. Since the (100) crystal plane exhibits higher reactivity under conventional etching conditions, its morphological evolution is significantly more sensitive to process parameters than other crystal orientations. Therefore, this experiment selected the (100) crystal plane as the research object to systematically investigate the synergistic effects of TMAH concentration gradients (5–25 wt%) and temperature (80 °C constant). Through cross-scale characterization using SEM and WLI, the variation patterns of roughness Ra at different concentrations can be quantified, revealing the interaction boundary conditions between crystal orientation and etching kinetics.

To explore the influence of etchant temperature on surface morphology, this study employed scanning electron microscopy (SEM, manufacturer: ZEISS of Germany, model: SPRA-55) and white light interferometry (WLI) to systematically characterize the surface morphology and roughness of silicon wafers etched in TMAH solutions of 5 wt%, 10 wt%, 15 wt%, 20 wt%, and 25 wt% at 80 °C [7]. As shown in Figure 6, the surface topography analysis demonstrates the evolution of the etched topography on the (100) crystal surface under different etchant concentration conditions, where the overall structural image, the high-magnification topography image, and the cross-section image are shown in order from top to bottom.

Experimental observations revealed that as the TMAH concentration increased from 5 wt% to 20 wt%, the density of surface hillocks gradually decreased, and the surface morphology became smoother [9,22]. At a concentration of 25 wt%, almost no hillock structures were observed, achieving the best surface smoothness [23,24]. This trend can be attributed to the uniform distribution and effective diffusion of OH^−^ ions in high-concentration TMAH solutions [12], which suppress localized etching and result in more uniform surface etching.

As observed in Figure 7a–e, the surface roughness of the (100) crystal plane exhibited a significant decrease with increasing [25] TMAH concentration. The minimum roughness (Ra = 35.37 nm) was achieved at 25 wt%, while the maximum roughness (Ra = 819.87 nm) was observed at 5 wt%. This phenomenon can be explained by the reaction mechanism: during the etching process between TMAH and silicon, unstable orthosilicic acid (H_4_SiO_4_) and hydrogen gas (H_2_) are generated [26,27,28]. Orthosilicic acid further decomposes to form metasilicic acid (H_2_SiO_3_) precipitates, which act as micro-masks on the silicon surface, while the generated hydrogen bubbles form pseudo-masks [29]. These micro-masks and pseudo-masks create localized shielding effects: the etching process is inhibited in covered regions, while exposed regions continue to undergo etching. This selective etching mechanism leads to the formation of numerous pyramidal hillock structures on the surface.

In order to obtain a flat and smooth sensitive film, we chose 25 wt% TMAH at 80 °C for sensitive film etching.

#### 3.1.2. Effect of Sensitive Membrane Roughness on Sensor Performance

Based on the experimental results, it can be concluded that during wet etching in an 80 °C water bath, a nearly smooth surface is formed in a 25 wt% TMAH solution, whereas in a 5 wt% TMAH solution, the sensitive membrane surface develops irregularly distributed pyramidal hillocks with heights ranging from 10 nm to 1 μm. These pyramidal hillocks will increase the thickness of the membrane, and the distribution of the pyramidal hillocks is not uniform, resulting in an uneven stress distribution, which causes the offset of the stress concentration area, and the full range output also changes. These changes will change the linearity, consistency, and repeatability of the sensor, which is not controllable. Due to the inherent randomness of etching morphology details, finite element simulations cannot accurately replicate the most realistic distribution of all hillocks. Consequently, existing studies on the electromechanical coupling simulation of pressure sensors typically assume an idealized smooth surface. To simulate the etching outcomes of both 5 wt% and 25 wt% TMAH solutions, a completely smooth surface is used to represent the morphology after 25 wt% TMAH etching, while a surface with hillocks covering 1/4 of the base area is employed to represent the morphology after 5 wt% TMAH etching, simulating the extreme case of irregular hillock distribution. The presence of localized hillocks results in localized increases in the thickness of the sensitive membrane.

As illustrated in Figure 8, the maximum stress for the smooth surface (25 wt% TMAH) is 138.257 MPa, whereas for the surface with hillocks (5 wt% TMAH), the maximum stress increases to 173.273 MPa, representing a 25.33% increase. The experimental design specifies a full-scale output of 100 mV and a sensitivity of 5 mV/V/MPa. The smooth surface yields a full-scale output of 101.03 mV and a sensitivity of 5.0515 mV/V/MPa, while the surface with hillocks produces a full-scale output of 109.162 mV and a sensitivity of 5.4581 mV/V/MPa. The simulation results demonstrate that when the sensitive membrane surface is smooth and free of hillocks, the full-scale output and sensitivity of the pressure sensor are closer to the design values.

During the mass production of pressure sensors, approximately 500 chips are fabricated on a 4-inch wafer. Due to variations in bump distribution and sensitive membrane roughness across individual chips, the stress distribution differs from chip to chip, resulting in performance disparities and poor consistency within a production batch. To address this issue, it is crucial to minimize the roughness of the sensitive membrane and enhance the consistency among chips in the same batch. In this study, a TMAH solution with 25 wt% was selected. As can be seen from Figure 7, at 25 wt%, the minimum surface roughness of the (100) crystal plane is Ra = 35.37 nm, while at 5 wt%, the maximum roughness is Ra = 819.87 nm. The difference between the two is more than 23 times, thus optimizing the uniformity and performance of the pressure sensor.

### 3.2. Manufacture of Pressure Sensors

In this study, piezoresistivity pressure sensor chips are fabricated in batches on a 4-inch silicon wafer using a five-times mask process. Figure 9 shows the steps involved in the fabrication of a piezoresistivity pressure sensor chip [21]. The process manufacturing details are as follows:

A 4-inch double-sided polished silicon wafer (100) with a thickness of 500 μm and a resistivity of 1–10 Ω cm is prepared.A thin layer of dioxide (300 nm) and a thin layer of nitride (300 nm), which are attached to both sides of the silicon wafer, are grown on the wafer by low-pressure chemical vapor deposition (LPCVD). The passivation layer is mainly used as a mask in the wet etching process.The oxide and nitride layers are dry etched by reactive ion etching (RIE), windowing the subsequent areas that require wet etching, and areas that do not require wet etching are still protected by the passivation layer.Taking advantage of the high selectivity of silicon for wet etching in alkaline solutions, it was etched in a 25% tetramethylammonium hydroxide (TMAH) solution for about 17 h, and the etching was stopped when the thickness of the sensitive film reached 95 μm.The silicon nitride layer on both sides of the wafer is precisely removed using a reactive ion etching process, followed by the removal of the silicon oxide layer on both sides of the wafer using a BOE solution at a constant temperature of 40 °C [25] in a water bath to ensure that all passivation layers are completely removed.Ion injection of boron into the device layer is performed to complete the doping of the entire surface of the device layer. After ion implantation, the device was annealed for 20 min at 1000 °C to repair lattice damage.The deep silicon etching technique is utilized to pattern the device layers and prepare the piezoresistors and electrical isolation tanks.To form good ohmic contact, besides heavily doping the semiconductor surface, it also requires that the metals have a matching work function. For this reason, we chose metal Au material to prepare metal thin films by magnetron sputtering and to make leads and pads by peeling (Ti 50 nm, Pt 50 nm, Au 500 nm).A thin oxide layer is deposited on top of the prepared device layer using low-pressure chemical vapor deposition to protect the device layer from oxidation and significantly improve the stability and reliability of the device.Dry etching of the oxide layer by reactive ion etching exposes the sputtered metal pads in h to provide a reliable connection interface for gold lead bonding in the subsequent packaging process.Absolute pressure reference chamber by anodic bonding of silicon and glass (temperature 400 °C, pressure 1250 N, voltage 800 V) is performed.Cross-section of the overall MEMS chip structure is performed.

## 4. Measurement and Discussion

In order to verify the accuracy of the actual test results with the finite element analysis results, the experiments were conducted on the pressure sensors fabricated at 80 °C and 25 wt% TMAH conditions, taking into account the actual fabrication requirements. The pads of the sensor chip were bonded to the terminals of the stainless steel housing using leads. This established the necessary electrical connection to the external circuit, which comprises two matching resistors designed for the chip, a power supply, and a device for measuring output voltage. The encapsulated sensor is linked to the pressure pump’s interface via a stainless steel pipe thread and gasket, enabling the pump to supply the requisite pressure conditions for calibration [15]. Figure 10 illustrates the complete testing system assembled in a laboratory setting.

The bench pressure pump is used as the pressure source (ConST162, manufacturer: Beijing ConST Instrument Technology Co., Ltd. in Beijing, China), the pressure sensor is calibrated by a pressure gauge fitted with the pressure pump (ConST273, manufacturer: Beijing ConST Instrument Technology Co., Ltd. in Beijing, China), and the power is supplied by a 5 V DC power supply (Lvyang YB1702, Green Yang Technology Co. LTD., Baoan District, Shenzhen, China). The output is measured by a high-precision multimeter (Agilent 34401A, Keysight Technologies, Palo Alto, CA, USA) [5]. The accuracy class of the above pressure gauge is ±0.1%FS.

The pressure calibration experiment calibrates the pressure between 0 kPa and 4 MPa at room temperature. The experimental results of the three-round journey are shown in Figure 11, which includes the line of the three rounds of test results. For a test range of 4 MPa, the sensitivity of the pressure sensor prepared without signal amplification is 5.071 mV/V/MPa, while the finite element simulation result of the designed structure is 5.0515 mV/V/MPa. The test results were only 0.39% higher than the simulation results, indicating a high degree of consistency between the experimental data and the simulated predictions. This close agreement confirms the reliability of both the sensor design and the simulation model.

The test results show that the pressure sensor has a sensitivity of 5.071 mV/V/MPa, a linearity of about 0.67%, a hysteresis of about 0.88%, and a repeatability of about 0.75%.

## 5. Conclusions

This study proposes a manufacturing method for silicon piezoresistive pressure sensors based on a simplified wet etching process. Through finite element analysis, the optimal chip size of 3200 μm × 3200 μm × 1000 μm was determined. Under optimized process parameters (temperature 80 °C, TMAH concentration 25 wt%), uniform etching of the diaphragm structure (1000 μm × 1000 μm × 0.095 μm) was achieved, with surface roughness as low as 35.37 nm. Compared to the traditional 5 wt% TMAH process, this method reduces stress non-uniformity by 95.7%. The process significantly reduces stress in homogeneity, improves repeatability and consistency, and enhances batch-to-batch stability by reducing plasma damage. The fabricated sensors exhibit excellent performance at room temperature: sensitivity of 5.071 mV/V/MPa, linearity of 0.67%, hysteresis of 0.88%, and repeatability of 0.75%. These comprehensive performance metrics surpass those of traditional dry etching processes.

## Figures and Tables

**Figure 1 micromachines-16-00516-f001:**
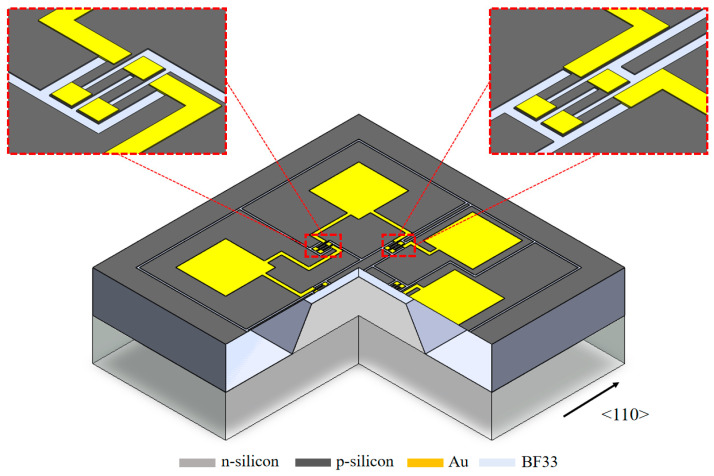
Schematic diagram of a pressure sensor.

**Figure 2 micromachines-16-00516-f002:**
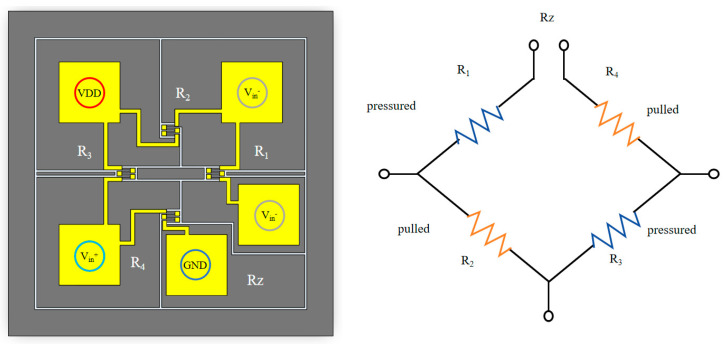
Equivalent circuit of a Wheatstone bridge.

**Figure 3 micromachines-16-00516-f003:**
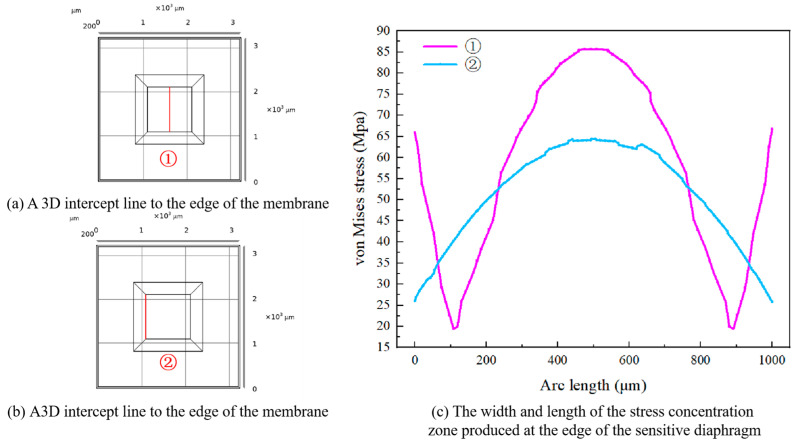
Path analysis of trinomial intercepts.

**Figure 4 micromachines-16-00516-f004:**
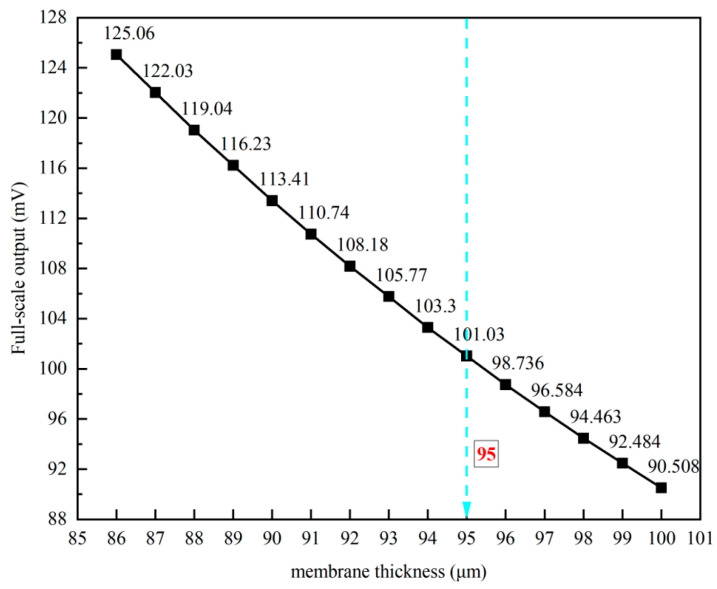
Simulation results of full-scale output versus diaphragm thickness.

**Figure 5 micromachines-16-00516-f005:**
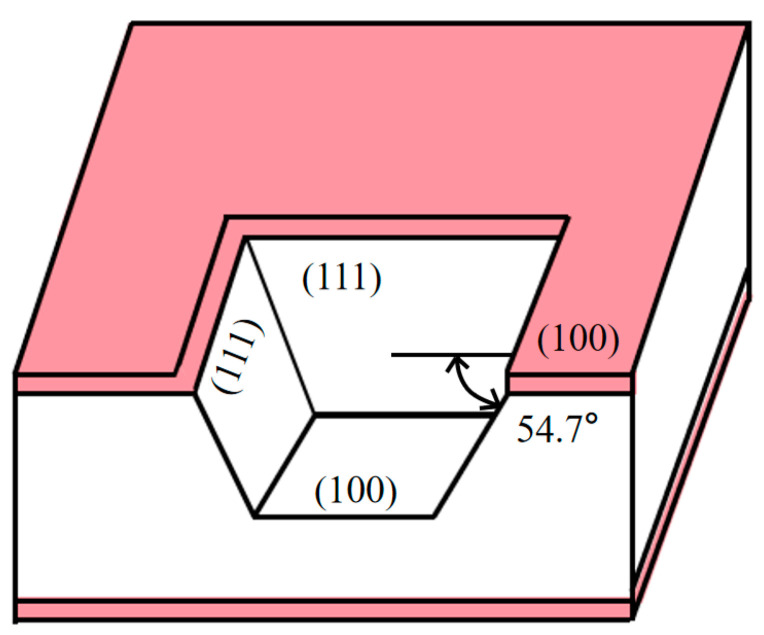
Structural view of the back cavity after wet etching.

**Figure 6 micromachines-16-00516-f006:**
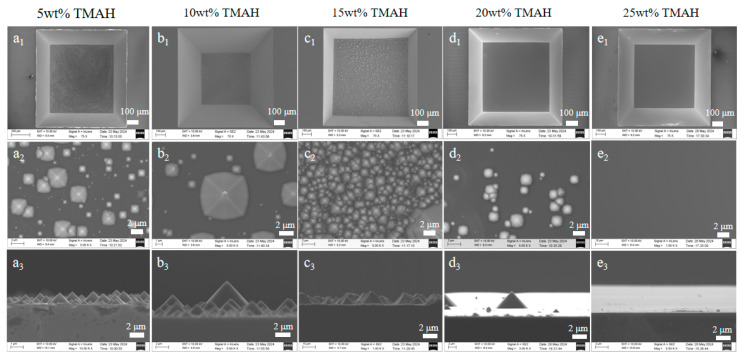
SEM image of Si (100) surface after etching with different concentrations of TMAH solution at 80 °C: (**a_l_**) overall top view with 5 wt% TMAH, (**a_2_**) high magnification top view with 5 wt% TMAH, and (**a_3_**) cross-section with 5 wt% TMAH; (**b_l_**) overall top view with 10 wt% TMAH, (**b_2_**) high magnification top view with 10 wt% TMAH, and (**b_3_**)cross-section with 10 wt% TMAH; (**c_l_**) overall topview with 15 wt% TMAH, (**c_2_**) high magnification top view with 15wt% TMAH, and (**c_3_**) cross-section with 15 wt% TMAH; (**d_l_**) overalltop view with 20 wt% TMAH, (**d_2_**) high magnification top view with20 wt% TMAH, and (**d_3_**) cross-section with 20 wt% TMAH; (**e_l_**)overall top view with 25 wt% TMAH, (**e_2_**) high magnification topview with 25 wt% TMAH, and (**e_3_**) cross-section with 25 wt% TMAH [7].

**Figure 7 micromachines-16-00516-f007:**
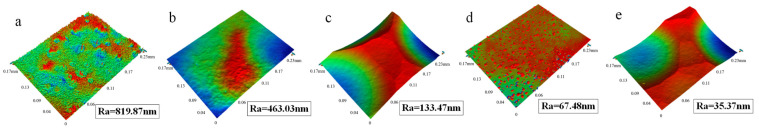
WLI plots of Si(100) surface after etching with different concentrations of TMAH solutions at 80 °C: (**a**) with 5 wt%TMAH, (**b**) with 10 wt% TMAH, (**c**) with 15 wt% TMAH; (**d**) with 20wt% TMAH; (**e**) with 25 wt% TMAH.

**Figure 8 micromachines-16-00516-f008:**
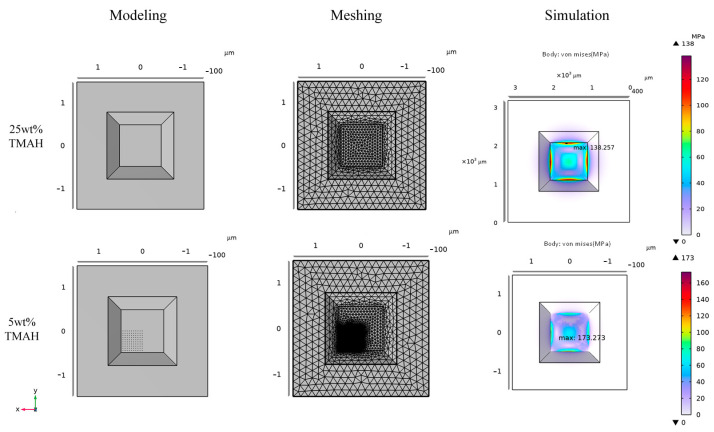
Finite element simulations of sensitive membranes with and without hillocks.

**Figure 9 micromachines-16-00516-f009:**
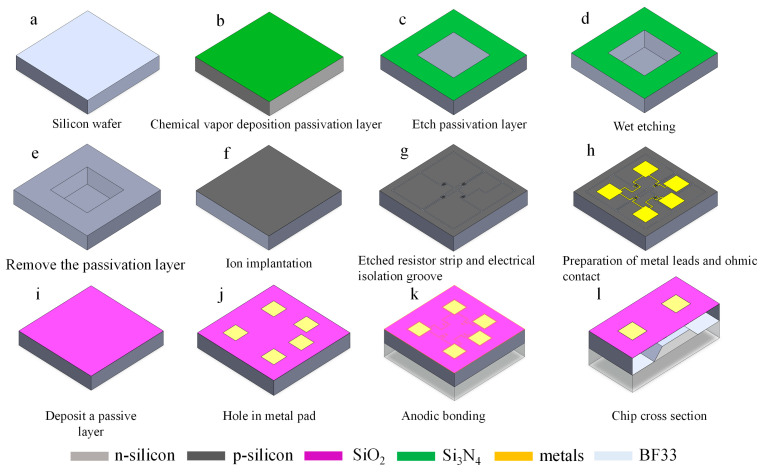
Process flow design diagram.

**Figure 10 micromachines-16-00516-f010:**
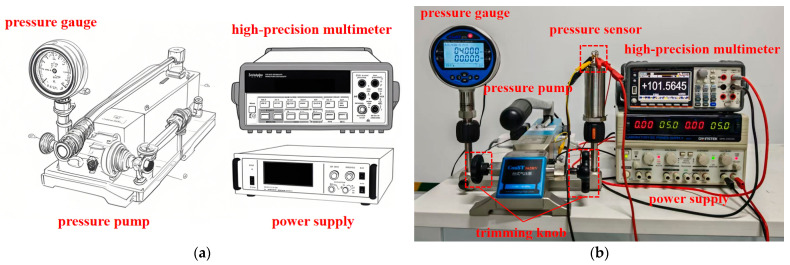
Pressure sensor calibration system: (**a**) structure diagram of test platform; (**b**) image of the test platform [30,31].

**Figure 11 micromachines-16-00516-f011:**
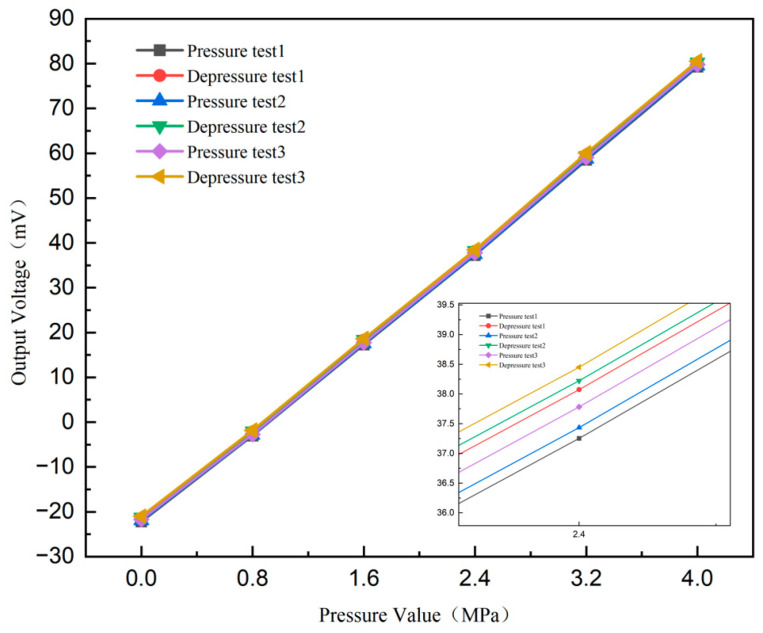
Normal temperature static pressure test chart.

## Data Availability

The original contributions presented in the study are included in the article; further inquiries can be directed to the corresponding author.

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
