# Peer review of "Design and Fabrication of Silicon Pressure Sensors Based on Wet Etching Technology"

_micromachines, 2025, doi:10.3390/mi16050516_

Round 1

Reviewer 1 Report

Comments and Suggestions for Authors

This paper presents a piezoresistive pressure sensor fabricated using wet etching process. However, the work demonstrates no novel contributions in sensor design, fabrication methodology, or performance metrics. Crucially, the study fails to clearly establish how the proposed wet etching technique enhances device performance compared to conventional approaches. Substantial revisions addressing the following points are required to meet publication standards.

1. Abstract: The authors stated "Simulation results demonstrate that pressure sensors with smoother sensitive membranes exhibit superior performance". The performance should describe accurately.
2. Line 37-39: Ease of miniaturization, and compatibility with integrated circuit (IC) fabrication processes would not contribute to the percison of pressure sensor.
3. Fig.2: What is R总?
4. Fig.3 How did the authors get the curves? based on theoretical analysis or simulation? What are the conditions?
5. Section 3.1.2 should carry out a comprehensive and in-depth analysis of the specific impact of membrane surface roughness on device performance, supported by detailed experimental data, to establish a strong correlation with the central theme of the article.
6. What is the accuracy level of pressure gauge?
7. I can't obtain any information from Figure 11. The curves all overlap each other. It is suggested to present it in a new way.
8. Pls carefully review the text. There are quite a few language errors such as:
Line 96 glass sheet should be glass cover;
Line 123 σm shoud be σm;
Line 139, what is "mh"?
Line 144 What is "Figure x"?
Figure 4, film should be membrane.

Reviewer 2 Report

Comments and Suggestions for Authors

Dear authors

I have overall enjoyed article reading. The topic discussed by the authors is interesting to the readers, and in general, the article is well written. I list below some major and minor changes that must be addressed before further article processing.

Major changes

Simulation results reported in Figure 3: Please provide more technical details regarding the simulation results from this figure, e.g.: what software was used for the simulation (include version)? What parameters were considered for this simulation (timestamp, mesh size, and so on)? Any other detail worth mentioning.

Figure 5 shows the orientation of the pressure sensor on the Silicon wafer. Could you please explain why was this orientation chosen for sensor fabrication? How would sensor response be modified in case of sensor rotation to a different orientation? Please discuss about it.

Overall paper organization: The paper is organized in a hard-to-read manner. Figure 6 (line 185) presents SEM images of the assembled sensors; however, the manufacturing process is presented later in line 237 and ahead; please re-organize the document in a more readable fashion.

Regarding conclusion section: Could you please discuss the pros and cons of wet etching technique compared with traditional manufacturing? please include the relevant experimental results from this study in the discussion; this would really enhance the quality of your manuscript.

Minor changes

Please make sure to define all symbols before using them in Equations (2) through (4); this observation holds for the symbols: wmax, E, A, B, h, Vin and Vout

Figure 3: This figure has 3 subfigures, but the caption fails to provide a description for each subfigure. Also increase font size for text in subfigures (a) and (b).

Figure 6 reports SEM images of Si (100) surface, please provide the details of SEM equipment (manufacturer and model).

Figure 7: axes’ names are barely readable in subfigures. Please enlarge them.

Round 2

Reviewer 1 Report

Comments and Suggestions for Authors

The authors have satisfactorily resolved all major technical and presentation issues identified during the initial review. This manuscript now meets the journal's publication standards. I recommend acceptance in its current form.

Reviewer 2 Report

Comments and Suggestions for Authors

Mandatoy changes have been replied one by one. The article can be published in present form.

For future articles, please elaborate the authors' response letter in a way that the reviewer concerns are followed by the authors' reply. This helps the reviewer to check that each concern has been replied individually.